# Application of Exogenous Ascorbic Acid Enhances Cold Tolerance in Tomato Seedlings through Molecular and Physiological Responses

**DOI:** 10.3390/ijms251810093

**Published:** 2024-09-19

**Authors:** Xinman Wang, Chunxia Ran, Yuandi Fu, Liyuan Han, Xuedong Yang, Weimin Zhu, Hui Zhang, Yingying Zhang

**Affiliations:** Shanghai Key Laboratory of Protected Horticultural Technology, Horticulture Research Institute, Shanghai Academy of Agricultural Sciences, Shanghai 201403, China; wangxinman@saas.sh.cn (X.W.); chunxiaran2020@163.com (C.R.); fuyuandi@sibcb.ac.cn (Y.F.); hanliyuan1234@outlook.com (L.H.); yxuedong@hotmail.com (X.Y.); yy17@saas.sh.cn (W.Z.)

**Keywords:** tomato, cold stress, ascorbic acid, seedlings, tolerance

## Abstract

Ascorbic acid (AsA), an essential non-enzymatic antioxidant in plants, regulates development growth and responses to abiotic and biotic stresses. However, research on AsA’s role in cold tolerance remains largely unknown. Here, our study uncovered the positive role of AsA in improving cold stress tolerance in tomato seedlings. Physiological analysis showed that AsA significantly enhanced the enzyme activity of the antioxidant defense system in tomato seedling leaves and increased the contents of proline, sugar, abscisic acid (ABA), and endogenous AsA. In addition, we found that AsA is able to protect the photosynthetic system of tomato seedlings, thereby relieving the declining rate of chlorophyll fluorescence parameters. qRT-PCR analysis indicated that AsA significantly increased the expression of genes encoding antioxidant enzymes and involved in AsA synthesis, ABA biosynthesis/signal transduction, and low-temperature responses in tomato. In conclusion, the application of exogenous AsA enhances cold stress tolerance in tomato seedlings through various molecular and physiological responses. This provides a theoretical foundation for exploring the regulatory mechanisms underlying cold tolerance in tomato and offers practical guidance for enhancing cold tolerance in tomato cultivation.

## 1. Introduction

Low-temperature (LT) stress is a significant abiotic factor that strongly inhibits plant growth and development, yield, and vegetation distribution [1]. To cope with cold stress, plants have developed intricate mechanisms involving signal recognition and transduction, transcriptional and metabolic alterations, and phytohormone homeostasis to modulate their tolerance [2]. Exposure to low temperatures can compromise cell membrane stability, leading to variations in lipid composition, changes in membrane structure, and the leakage of cellular contents. Furthermore, low temperature also induces the overproduction of reactive oxygen species (ROS), resulting in oxidative damage [3,4]. Initially, the cold signal is firstly perceived by membrane-localized proteins, which then triggers Ca^2+^ influx, initiating downstream pathways in the cold stress signaling pathway [5,6]. Plants enhance their cold tolerance by synthesizing various chemicals and functional proteins, such as soluble sugar, proline, and cold-responsive proteins. These substances regulate multiple physiological processes under cold stress, including cell membrane stability, osmotic potential balance, and reactive oxygen species (ROS) scavenging in plants [7,8].

The components of cold stress-triggered signaling components involve messenger molecules, protein kinases, phosphatases, and transcription factors, which constitute an elaborate regulatory network to enhance plant cold tolerance [9,10]. The inducer of CBF expression1 (ICE1), C-repeat binding factor/dehydration-responsive element binding (CBF/DREBs), and cold-regulated (COR) constitute the well-known ICE1-CBF/DREBs-COR cascade, which has been extensively reported as the most important cold-regulated pathway [10,11]. ICE binds to the typical MYC *cis*-elements (CANNTG) in the promoters of *CBF1/2/3*, thereby promoting the expression of *CBF1/2/3* under cold stress to enable plants to better cope with cold stress [12]. The activity and stability of ICE1, a key positive regulator in the cold signaling cascade, are modulated by various factors including E3 ubiquitin ligase SIZ1 and HOS1, MPK3, MPK6, and OST1 [9,13,14]. Recent research has unveiled that the SlGRAS4-related signaling pathway modulates cold response independently of the ICE1-CBF pathway in tomato. SlGRAS4 directly regulates *SlCBF1/2/3* expression by binding to their promoters. In addition, SlGRAS4 can activate the expression of other target genes involved in antioxidant capacity, calcium signaling, photosynthetic capacity, and energy metabolism [15].

AsA is an important non-enzymatic antioxidant in plants, crucial for plant growth, development, and antioxidant functions [16]. AsA plays vital roles in various processes including photoprotection, the wounding response, and ethylene and gibberellin biosynthesis. Moreover, AsA regulates cell division and expansion, thereby controlling cell growth [17,18]. AsA influences the activity of antioxidant enzymes such as SOD, POD, CAT, and APX and enables the production of non-enzymatic antioxidants, thereby protecting plants against oxidative stress-related damage through effectively scavenging ROS [19]. The application of AsA enhances photosynthetic capacity and maintains ion homeostasis in wheat under salt-induced oxidative stress, mitigating the adverse effects of salt stress on wheat growth [20,21]. The ABI4-VTC2 module regulates seedling growth under salt stress by modulating AsA levels to scavenge ROS accumulation [22]. Exogenous AsA may dramatically increase resistance to salt stress in many plants, including maize, rice, and alfalfa [23,24,25]. However, whether AsA modulates plant LT stress remains largely to be elucidated.

Cold stress leads to enormous yield losses due to reduced leaf area, photosynthetic efficiency, and plant biomass [26]. The degree of cold-induced damage in plants can be characterized by the appearance, temperature, and optical properties of the plants. Hyperspectral image analysis techniques, including RGB, multispectral, hyperspectral, thermal, chlorophyll fluorescence, and 3D sensors, have been increasingly applied to detect and differentiate early plant diseases and identify subtle changes in plant growth and development. These techniques offer the advantages of simple, rapid, and reagent-free [27,28]. One of the widely used metrics is the Normalized Difference Vegetation Index (NDVI), which is calculated as a simple ratio of near-infrared to visible light and is used to describe the physiological attributes of vegetation and thus represents overall health [29,30]. Chlorophyll content is an indicator of photosynthetic activity, stress, and nutritional status in plants. Studies using the ground truth data (chlorophyll content and reflectance data) of winter wheat during its growing season have confirmed a strong linear relationship between the Modified Chlorophyll Absorption Ratio Index (MCARI) and chlorophyll content [31]. Remote sensing plant stress indicators based on visible and near-infrared spectral regions offer an alternative approach to traditional plant stress parameter measurements. This method is currently widely applied in research on various crops such as wheat, rice, and tomato [32,33,34].

Tomato (*Solanum lycopersicum* L.) originates from warm tropical areas, and most tomato varieties are sensitive to low temperatures (0–15 °C) at various stages [35]. Exogenous AsA, known for its protective effects, is frequently employed in research on plant stress tolerance. However, the role of exogenous AsA in the cold tolerance of tomato seedlings has rarely been explored. The present study aimed to investigate the role of AsA in regulating cold stress tolerance and the underlying mechanisms in tomato seedlings. Our findings convincingly demonstrates that exogenous AsA positively improves cold tolerance in tomato seedlings. We propose that these results offer valuable insights for practical applications in tomato production and provide a theoretical basis for the use of exogenous AsA to enhance cold tolerance in tomato.

## 2. Results

### 2.1. Effects of AsA on the Growth and Development of Tomato Seedlings

Exogenous AsA treatment significantly alleviated the damage caused by cold stress in tomato, both morphologically and in terms of biomass distribution (Figure 1). Compared to the control (CK) group, the chilling (CK + L) group treated with LT for 7 days exhibited stunted seedling growth. The leaves were thin, accumulated a substantial amount of anthocyanin, and showed clear spots due to cold damage. Additionally, the leaf margins showed severe dehydration, wilting, and curling, suggesting significantly poorer overall growth. In contrast, the AsA + L group exhibited less leaf wilting and displayed better overall growth (Figure 1A).

Further observations revealed that the survival rate of seedlings in the AsA + L group was significantly higher than that in the CK + L group, with the survival rate in the AsA + L group being 6.5 times that of the CK + L group (Figure 1B). Under cold stress, the exogenous application of AsA resulted in a 25% increase in root length compared to the control group (Figure 1C). These results suggest that exogenous AsA treatment can effectively mitigate the adverse effects of LT on both the aerial and underground tissues of tomato seedlings, maintaining their morphological structure and improving cold tolerance.

### 2.2. Effect of Exogenous AsA on Chlorophyll Content and Photosynthesis of Tomato Seedlings

Chl fluorescence phenotypic investigation showed that tomato leaves in the CK + L group were considerably damaged, which was more severe in tomato leaves that were treated with cold stress for longer (Figure 2A). Under cold stress, the activity of PSII in leaves decreased, and the fluorescence parameters Fv/Fm, ETR, and qP continued to decrease, with maximum decreases of 28.6%, 58.5%, and 53.3%, respectively. The application of AsA significantly suppressed the cold-induced damage in tomato leaves and alleviated the cold-triggered decreases in Fv/Fm, ETR, and qP (Figure 2B–D). Furthermore, we found that exogenous AsA treatment significantly increased the NPQ value in tomato under cold stress, increasing by 10.7–51.9% (Figure 2E).

Under normal growth conditions, AsA pretreatment displayed no significant effect on the content of Chl-a but increased the contents of Chl-b and Chl-c by 73.4% and 20.9%, respectively. Under LT conditions, the total contents of Chl-a, Chl-b, and Chl-c in the leaves decreased significantly. However, the AsA + L group exhibited a slower rate of decrease, and the Chl contents were nearly two times higher than those in the CK + L group (Figure 2F–H).

We then identified the expression level of the ribulose-1, 5-bisphosphate carboxylase/oxygenase gene (*SlRubisco*), which is a key photosynthetic carbon fixation enzyme gene in plants [36]. Before cold treatment, the exogenous application of AsA increased the expression of *SlRubisco*. After cold treatment for 1 h, the expression of *SlRubisco* significantly decreased by 76.8%, but its expression level in the AsA + L group remained higher than that in the CK + L group, suggesting that exogenous AsA could mitigate the decrease in *SlRubisco* expression.

### 2.3. Effects of Exogenous AsA on High-Flux-Related Plant Phenotypes of Tomato Seedlings under Cold Stress

As illustrated, no significant differences were found in VIs among the groups without cold treatment (Figure 3). After 7 days of cold treatment, the RE-NDVI, GNDVI, VOG1, and CIgreen indices, which positively correlate with chlorophyll content and vegetation health, significantly decreased in the tomato seedlings subjected to cold stress. The CK + L group decreased by 29.1%, 13.8%, 13.2%, and 25%, respectively, and the AsA + L group decreased by 25%, 5.28%, 10.6%, and 0.5%, respectively. The expression values of the CK + L group were significantly lower than those of the AsA + L group, which were 12%, 7%, 9%, and 43%, respectively (Figure 3A–D). Conversely, the MCARI, which negatively correlates with chlorophyll content and vegetation health, increased 11% in the AsA + L group compared with that in the CK + L group (Figure 3E). These findings suggest that cold treatment significantly decreased the chlorophyll content and growth in tomato seedlings. Interestingly, AsA treatment decreased cold-induced decrease in chlorophyll content and overall health status. These results demonstrates that although AsA cannot completely eliminate the damage caused by cold stress on tomato seedlings, it can significantly mitigate cold-induced adverse effects.

### 2.4. Exogenous AsA Enhanced the Antioxidant Capacity of Tomato Seedlings

To further investigate the effects of exogenous AsA on ROS metabolism in tomato leaves under LT, we analyzed the accumulation levels of O^2−^ and H_2_O_2_ and the activity of POD and SOD in the leaves. Interestingly, both the results of DAB staining and O^2−^ and H_2_O_2_ concentrations showed that during LT treatment, the CK + L group exhibited more brown areas and severe oxidative damage in tomato leaves (Figure 4A–C). Compared to the CK group, the contents of O^2−^ and hydrogen peroxide in the AsA group were slightly higher, with no significance. However, after 7 days of LT treatment, the accumulation levels of O^2−^ and hydrogen peroxide in the CK + L group were higher than that in the AsA + L group by 3.4 and 1.28 times of detection, respectively (Figure 4B,C). Under normal growth and LT conditions, the MDA content in AsA-treated tomato leaves did not significantly change. However, under cold conditions, the MDA content in AsA-treated tomato leaves was significantly lower than that in the mock-treated group (Figure 4G). Exogenous AsA increased the activities of POD and CAT by 89% and 46.3%, respectively, in untreated tomato seedings, but there was a slight increase in SOD activity. After 7 days of LT treatment, the activities of POD, SOD, and CAT in the AsA + L group were more than twice those in the CK + L group (Figure 4B–D). Our findings revealed that AsA treatment contributed to the scavenging of ROS in tomato seedlings.

Under LT conditions, the endogenous AsA content in tomato seedlings significantly increased, along with the increased expression of the key AsA biosynthesis gene *SlGLDH*. After exogenous AsA treatment, the accumulation of endogenous AsA content in tomato plants was obviously increased by 125.37 ug·g^−1^ (AsA) and 243.05 ug·g^−1^ (AsA + L), respectively (Figure 4H). Moreover, compared to untreated plants, the expression level of *SlGLDH* was higher after AsA application. AsA pretreatment may act as an exogenous signal to induce the expression of *SlGLDH* in tomato plants (Figure 4I). These results suggested that exogenous AsA may initiate the synthesis of endogenous AsA in plants, potentially through a preventive mechanism.

### 2.5. Exogenous AsA Promoted the Accumulation of Soluble Sugars, Proline, and ABA in Tomato Seedlings

We further investigated the changes in two well-known stress defense substances, proline and sugars, in tomato leaves under different treatments. Under normal conditions, exogenous AsA increased the concentrations of sugars and proline in tomato leaves but displayed no obvious effect on sucrose content. After LT treatment, the sucrose content in the tomato leaves of the CK + L group significantly decreased, while exogenous AsA increased the sucrose content by 84.7%, proline content by 75.6%, and soluble sugar content by 48.9% (Figure 5). These results suggest that AsA may help tomato seedings adapt to a low-temperature environment by promoting the biosynthesis of osmotic regulatory compounds, such as soluble sugars, sucrose, and proline.

Under LT conditions, the endogenous ABA content in tomato leaves of the AsA + L group significantly increased from 70 ng·mL^−1^ to 131.2 ng·mL^−1^, with an 87.5% increase. In contrast, the endogenous ABA content in the CK + L group increased from 55.6 ng·mL^−1^ to 96.6 ng·mL^−1^, with a 73.8% increase. As the duration of LT increased, the expression of the ABA signal transduction gene *SlABI3* (*ABA-insensitive3*, an ABA-dependent transcription factor) showed a significant increase in the CK + L group. However, the expression of the *SlABA3* (*ABA deficient 3*, an ABA synthesis gene) showed no significant changes.

Compared to the CK + L group, the expression levels of *SlABA3* and *SlABI3* showed significant increases in the AsA + L group upon LT treatment for 1 h (Figure 5E,F). The expression level of *SlABA3* in the AsA + L group decreased after prolonged low-temperature treatment, which may be due to the significant increase in endogenous ABA levels and the elevated background level of *SlABA3* itself following AsA pretreatment.

### 2.6. Exogenous AsA Afftected the Expression of Antioxidant-Related and Cold-Regulated Genes

We evaluated the expression levels of several genes, including antioxidant-related genes *SlPOD*, *SlSOD*, *SlCAT*, *SlAPX*, *SlGPX,* and *SlGlut*, as well as *SlICE1*, *SlGRAS4*, *SlCBF1*, *SlCBF2,* and *SlCBF3* involved in the cold stress pathway. These genes have all been reported to play important roles in tomato responses to cold stress (Figure 6). After cold treatment, the expression levels of the *SlPOD*, *SlCAT*, *SlAPX*, and *SlGPX* genes in the control group significantly increased. *SlPOD* exhibited an upregulation of 45.2 times that of the control group and reached its peak within 1 h of cold exposure (Figure 6A).

The expression of *SlAPX* and *SlGPX* was upregulated at 12 h of LT. These results indicated that cold stress can induce the expression of these genes, and in the short term, *SlPOD* may play an important role in the antioxidant enzyme pathway under cold treatment. Compared to the control, spraying AsA altered the baseline expression levels (0 h) of antioxidant-related genes in tomato plants, with significant upregulation of the expression levels of *SlPOD*, *SlSOD*, *SlAPX*, and *SlGPX*. Notably, under LT conditions, the expression trends of these antioxidant genes were changed, except for *SlPOD*, *SlSOD*, and *SlCAT*. Compared to the control group, the expression levels of *SlPOD* and *SlCAT* further increased after 1 h of cold exposure (Figure 6A–C). The expression trends of *SlSOD*, *SlAPX*, and *SlGPX* in the AsA + L group differed from those in the CK + L group, but their expression levels were higher than those in the CK + L group (Figure 6D,E). Interestingly, the background level of the *SlGlut* gene in tomato plants did not increase as the genes mentioned above after the external application of AsA, but its expression was rapidly upregulated after 12 h of LT, which was higher than that in the CK + L group. Its expression was upregulated by 313.3% (Figure 6F).

We then analyzed the expression of several key genes involved in the ICE1-CBF pathway and the SlGRAS4 pathway (Figure 7), two currently recognized tomato cold-regulated pathways. The results demonstrated that after 1 h, the expression of *SlICE1*, *SlCBF1*, *SlCBF2*, and *SlCBF3* was dramatically upregulated. Compared to the CK + L group, exogenous AsA treatment further upregulated the expression of these genes (Figure 7A,C,D). However, the expression trend of *SlGRAS4* in the exogenous AsA-treated group was similar to that in the control group and did not display substantial changes (Figure 7E). These data suggest that the effect of AsA in promoting cold tolerance primarily operates through the ICE1-CBF pathway.

## 3. Materials and Methods

### 3.1. Plant Materials and Growth Conditions

The material NRP20-7-2 used in this experiment is a cold-sensitive species obtained from the Horticultural Research Institute, Shanghai Academy of Agricultural Sciences. Seeds were routinely sterilized and sown in 72-well seedling trays; after reaching the two-true-leaf stage, the seedlings were kept in a climate chamber under a 16:8 h light/dark photoperiod with a temperature of 25 °C/20 °C (day/night) and 300 μM·m^−2s^·^−1^ of light intensity, where the relative humidity was 60–70% [25].

### 3.2. Treatments

Tomato seedlings of 30 days old were divided into four groups: control (CK), AsA, chilling (CK + L), and chilling + AsA (AsA + L). The environmental conditions were as follows: The CK and AsA groups were subjected to normal conditions, while the other two groups were transferred to a room at 4 °C. Based on previous studies, a 10 mM AsA solution was selected to pretreat the tomato seedlings. Foliar spraying with distilled water was applied to the CK and CK + L groups until runoff occurred (Appendix A). Spraying began at 9:30 a.m., ensuring that each plant received the same volume of AsA solution or distilled water, and then the plants were kept at 25 °C for 24 h after spraying was completed. Subsequently, the seedlings were subjected to their respective treatments for 7 days. Each treatment was replicated three times, with 72 individual plants per replicate. On days 0, 1, 3, 5, and 7 of the treatment, the third fully expanded leaves were collected for 3,3-diaminobenzidine (DAB) staining and fluorescence measurements. Each assay was repeated three times.

### 3.3. Morphological Observation and Root Length Determination

To examine the phenotypes of tomato seedlings, we photographed seedlings after 7 days of different treatments and measured root length (below the root neck) and survival rate for each treatment in 72 independent biological replicates.
$$Plant\;survival\;rate=\frac{N_{plant\;survived}}{N_{plant}}\;\times \;100\%$$

### 3.4. Chl Fluorescence Parameters and Chlorophyll Content

To measure the chlorophyll (Chl) fluorescence parameters, we used the Chl fluorescence imaging system (WALZ, Echterdingen, Germany, IMAG-MAX/L). Seedlings were dark-adapted for 20 min. First, the leaf surface was exposed to modulated measuring light (0.6 kHz, PPFD ≤ 0.1 μmol m^−2^·s^−1^, “weak red light”) to measure initial fluorescence (Fo). Then, the saturation pulse light (20 kHz, 300 ms pulse of 10,000 μmol m^−2^ s^−1^, “white light”) was applied to determine maximal fluorescence (Fm) and the maximal change in P700. Next, actinic light (AL, 531 μmol m^−2^·s^−1^) was used to stimulate normal photosynthesis for several minutes. During illumination, steady-state fluorescence (Fs) and maximal fluorescence in this light (Fm) were obtained. The tomato seedling leaves were darkened for 20 min before being measured [37]. Winterman’s and De Mot’s approach was used to determine the Chl content [38]. For each treatment, each leaf was selected at three points on the same part of each leaf, and the main leaf veins were avoided.

### 3.5. DAB Chemical Staining 

DAB dye is often used to assess the level of peroxide and superoxide in fresh leaves. Parts of the leaf of tomato plants were taken for DAB staining, with a DAB solution concentration of 1 mg/mL and a pH of 3.8. The removed leaves were immediately placed in DAB solution, followed by vacuuming until the leaves sank to the bottom, after which the leaves were put in an incubator at 28 °C for 6–10 h [39]. The process was accompanied by the precipitation of dark red material, and the leaves were placed in 95% ethanol, boiling the water bath for 10 min, repeatedly 3 times, and then decolored gradually in 85%, 70%, and 50% ethanol solutions in turn; finally, they were photographed with a camera (EOS M6, Tokyo, Japan).

### 3.6. Determination of Physiological Indicators

To assess the degree of cold injury to tomato leaves, we examined the MAD content, H_2_O_2_ content, O^2−^ content, proline content, soluble sugar content, sucrose content, POD, SOD, CAT activity, and endogenous AsA level of the third completely developed leaf on tomato plants after 7 days of treatment. The MDA content was measured using an MDA reagent kit (Suzhou Comin Biotechnology Co., Ltd., Suzhou, China). The production of a superoxide radical (O^2−^) was determined using an O^2−^ reagent kit (Suzhou Comin Biotechnology Co., Ltd., Suzhou, China). Hydrogen peroxide (H_2_O_2_) was measured using a H_2_O_2_ reagent kit (Suzhou Comin Biotechnology Co., Ltd., Suzhou, China). The indices listed above were determined to use an enzyme marker (INFINITE 200 PRO, TECAN, Männedorf, Switzerland). The calculated approach of the aforementioned indices refers to Chen et al. [40]. According to the instructions, the ABA content of leaves treated for 7 days (three independent biological replicates) was determined (ELISA, Shanghai, China).

### 3.7. Hyperspectral Data Acquisition and Processing

In a darkroom setting, hyperspectral data were gathered, processed, and analyzed for tomato seedlings arranged on a black light-absorbing backdrop. A hyperspectral imaging system was employed to capture hyperspectral images of the seedlings, consisting of a hyperspectral imaging instrument, a halogen light source, and an electric drive platform. The hyperspectral imager (Resonon Pika L, LICA United Technology Limited, Beijing, China) was a push-broom type device with a spectral range of 400–1000 nm, a spectral resolution of 2.1 nm, 281 spectral channels, and 900 spatial channels. ENVI 5.5 software was utilized to extract each tomato seedling area from the background in the hyperspectral image using a decision tree classification method. Additionally, the chemical properties of the plants were quantitatively assessed at the individual plant level. The average spectral reflectance data for all pixels in the pure tomato seedling image area were calculated as the spectral reflectance data. Considering the increased noise at the beginning and end wavelengths of the equipment, further analysis was performed on the spectral reflectance within the 410–850 nm range.

The spectral reflectance of each tomato seedling was used to calculate five vegetation indices (VIs) which were closely related to the plant’s physiology. RE-NDVI is sensitive to canopy foliage content and senescence and could be used to detect vegetation stress [41]. GNDVI is sensitive to chlorophyll concentration and related to the radiation proportion absorbed photosynthetically [42]. VOG1 is sensitive to the combined effects of foliage chlorophyll concentration, canopy leaf area, and water content [43]. The MCARI is highly sensitive to both chlorophyll concentrations and variations in leaf area index and could be used to assess the depth of chlorophyll absorption and its concentration variations [31,44]. The total amount of chlorophyll in plants could be calculated using CIgreen [45]. NDVI, GNDVI, VOG1, and CIgreen positively correlate with chlorophyll content and vegetation health [41,42,43,45], whereas the MCARI exhibits negative correlations [31,44].

### 3.8. RNA Extraction and Expression Analysis

Leaves (the 3rd fully expanded leaf) at 0 h, 1 h, and 12 h of LT were used to obtain total RNA. The RNA extraction process followed our laboratory’s experimental protocol [46]. We examined the expression of representative genes. Appendix A lists the primers used for qPCR.

### 3.9. Statistics and Analysis

Data were produced using Microsoft Excel 2010 and analyzed for significance (one-way ANOVA and Duncan’s test) using the SPSS 20.0 package and plotted using GraphPad Prism 9 software.

## 4. Discussion

Low temperatures severely restrict the growth and productivity of field crops. Prolonged exposure to chilling stress negatively affects plant physiology, leading to leaf damage, stunted growth, decreased leaf area, wilting, and even death [47]. Exogenous AsA soaking seeds displayed an increased germination rate and seedling traits possibly through attenuating the negative effects of low-temperature stress on seed germination [48]. In this study, exogenous AsA can protect root development, leaf morphology, and plant growth, significantly alleviating the harmful effects induced by LT in tomato seedlings. In recent years, high-throughput plant phenotyping methods based on image data analysis have been employed to investigate plant responses to various biotic and abiotic stresses [32,33]. Hyperspectral phenotypic measurements of tomatoes indicate that the RE-NDVI, GNDVI, VOG1, and CIgreen index, which are positively correlated with chlorophyll content and plant health, were significantly higher in the AsA + L group compared to the CK + L group. Moreover, the MCARI, which is negatively correlated with chlorophyll content and plant health, was significantly higher in the CK + L group than that in the AsA + L group. Our results suggest that exogenous AsA provides a protective effect on tomatoes under LT stress.

Photosynthesis is often one of the major processes challenged by cold stress, which often hinders plants’ abilities to convert and use carbon dioxide [49]. Plants have evolved various defense mechanisms to mitigate stress-related ROS damage and photo inhibition. These mechanisms include switching between PSI and PSII states, altering the light-harvesting efficiency of PSII, dissipating excess thermal energy in PSII through non-photochemical quenching (NPQ), and fine-tuning the balance of electron transfer between PSI and PSII [4,50]. Our results indicated that exogenous AsA treatment can significantly mitigate the damage caused by low temperatures to the photosynthetic system of tomato. We discovered that under LT, the Chl-a, Chl-b, and Chl-total contents of tomato leaves were reduced. The content of Chl, as an important photosynthetic pigment involved in the modulation of the photosynthetic capacity of plants, was often affected by LT [51]. Further research revealed that the exogenous application of AsA can effectively reduce the loss of chlorophyll content in tomato leaves. At the gene level, the cold-induced downregulation of the key carbon-fixing enzyme gene *SlRubiSco* involved in photosynthesis was significantly mitigated in the AsA treatment group, further suggesting that AsA pretreatment can protect the photosynthetic system of tomato seedlings. Therefore, the regulation of chlorophyll content and photosynthesis may be another strategy by which AsA helps plants adapt to LT stress.

Under LT stress, the cell membrane becomes more permeable and unsaturated fatty acids accumulate, which lead to the formation of ROS and cause the buildup of membrane lipid peroxidation; MDA is the result of the breakdown of cell membrane lipid peroxidation [52]. Under abiotic stress, plants employ both enzymatic and non-enzymatic antioxidant mechanisms to remove ROS and alleviate oxidative stress [53]. SOD is an antioxidant enzyme that protect plants against ROS-related damage; POD and CAT catalyze the metabolism of H_2_O_2_ to simple water molecules by POD and CAT. In this study, without the addition of AsA, tomato leaves suffered extensive membrane lipid peroxidation damage, with significant accumulations of MDA, H_2_O_2_, and O^2−^ (Figure 4E–G). After the application of exogenous AsA, the activities of SOD and POD increased, and the accumulation of O^2−^ and H_2_O_2_ was significantly reduced. Other enzymes, such as ascorbate peroxidase and glutathione peroxidase, also play roles in scavenging H_2_O_2_ [54]. The qRT-PCR results further demonstrated that exogenous AsA decreased ROS generation in tomato leaves by activating antioxidant enzymes. As one of the most common non-enzymatic antioxidants in plants, AsA has been reported to maintain the internal cellular environment and protect plants under adverse conditions [17,55]. It is noteworthy that exogenous AsA elevated endogenous AsA levels and enhanced the antioxidant system of tomato plants, thereby reducing the damage caused by LT.

The main mechanisms underlying plant cold tolerance are also associated with alterations in sugar synthesis, protein metabolism, and the accumulation of proline [56]. In this study, a significant increase in proline accumulation was observed in tomato leaves (Figure 5C). Numerous studies revealed a positive correlation between proline accumulation and plant stress resistance. Proline acts as an osmolyte defense molecule and contributes to intracellular homeostasis, water absorption, osmoregulation, redox balance, the protection of cell structure integrity, and the stabilization of oxidative enzymes [57]. Soluble sugars contribute to membrane freezing resistance and provide an energy source for plants to adapt to low-temperature stress [58]. In our experiments, AsA application significantly increased the soluble sugar and sucrose content in tomato seedlings, suggesting that AsA enhanced the osmotic protection capacity of plants to some extent.

Plant hormones not only regulate growth and development by modulating various cellular, physiological, and developmental processes but also play a crucial role in plant stress responses [59,60,61]. The ABA signaling system has been identified as a key regulator of plant responses to abiotic stress, inducing significant changes in gene expression and adaptive physiological responses. LT stress induced ABA synthesis in tomato plants and increased the level of endogenous ABA. Intriguingly, AsA treatment initiated a stronger accumulation of endogenous ABA under LT stress (Figure 5D). The impact of exogenous AsA on endogenous ABA levels may result from a potential interaction between AsA and phytohormones [62]. AsA application enhances cold tolerance in tomatoes at the hormonal level, with an interaction between AsA and phytohormones that likely regulates ROS balance and subsequent developmental responses in plants.

Quite a few studies have demonstrated that under cold stress, many proteins were misfolded, leading to an overproduction of ROS, which harms plants. To defend themselves from the cold, plants produce cold-related protective proteins. ICE1 binds to the CBF cis-element to stimulate the expression of downstream *COR* genes, thereby enhancing plant cold tolerance [6,63]. At the gene level, exogenous AsA treatment increased the expression of positively regulated genes *SlICE1*, *SlCBF1*, *SlCBF3*, and *SlGRAS4* in the cold response pathway. These results further validate that AsA treatment enhances cold tolerance in tomato plants. Our results are consistent with Kosová’s findings that the tolerant genotype could successfully regulate energy metabolism to meet increased demands during stress adaptation, and that levels of several stress-related proteins (protective proteins, chaperone proteins, ROS scavenging, and detoxification-related enzymes) were significantly higher in the tolerant genotype compared to the sensitive genotype [64]. Given these insights, future applications in production could involve the use of exogenous AsA to improve cold tolerance in tomato cultivation. Furthermore, integrating this approach with other agronomic practices may optimize overall plant health and productivity, potentially leading to higher yields and improved fruit quality during adverse climatic conditions.

## 5. Conclusions

Plants are often subjected to extreme temperatures, including cold stress. Plants have developed a set of sophisticated mechanisms that allow them to withstand LT stress. Using certain exogenous cryoprotectants to protect crops against cold stress is one effective method in actual agricultural production. This study found that exogenous AsA plays a significant role in a multifaceted and coordinated network of responses to LT in tomato, including enhanced photosynthesis, antioxidant processes, and ABA synthesis; it also upregulated the expression of genes related to cold stress pathways and ABA synthesis pathways, accumulated some key metabolites, and increased antioxidant enzyme activity (Figure 8). In conclusion, our study provides novel mechanisms underlying exogenous ascorbic acid (AsA)-induced enhanced cold tolerance in tomato plants. The application of exogenous substances provides new thoughts and effective means to help plants withstand cold stress in agricultural cultivation. 

## Figures and Tables

**Figure 1 ijms-25-10093-f001:**
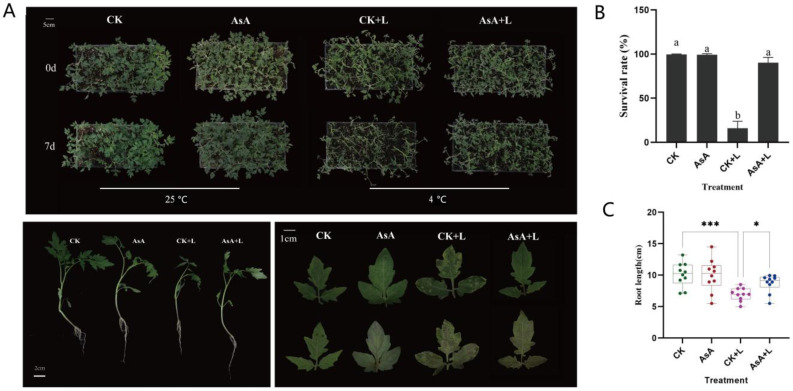
Effects of exogenous AsA on the photosynthetic system of tomato leaves under LT. (**A**) Images of tomato seedling phenotype. (**B**) Survival rate of seedlings at day 7. (**C**) Seedling root length at day 7 (* and *** represent significant at 0.05 and 0.001 level, respectively. Data are presented as the means of 3 biological replicates; different letters indicate significant difference at *p*-value < 0.05 using one-way ANOVA test).

**Figure 2 ijms-25-10093-f002:**
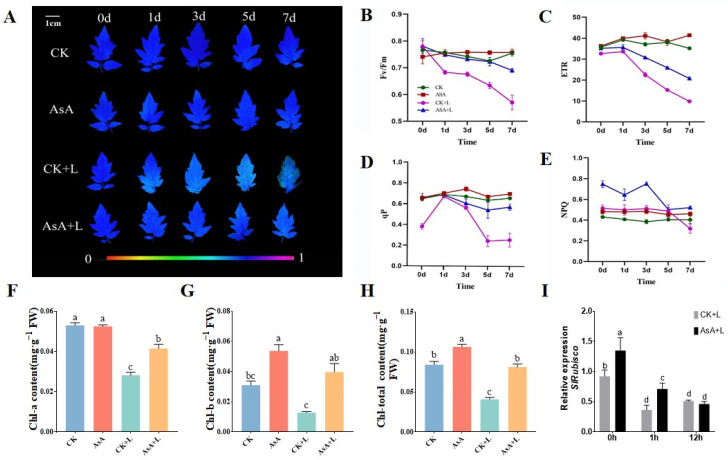
Effects of exogenous AsA on the photosynthetic system of tomato leaves under LT. (**A**) Images of chlorophyll fluorescence. (**B**) Maximum photochemical efficiency of PSII Fv/Fm. (**C**) Apparent photosynthetic electron transport rate (ETR). (**D**) Photochemical quenching coefficient (qP). (**E**) Non-photochemical quenching coefficient (NPQ). (**F**) Chl-a content. (**G**) Chl-b content. (**H**) Chl-total content. (**I**) *SlRubisco* expression level. Data are presented as the means of 3 biological replicates; different letters indicate significant difference at *p*-value < 0.05 using one-way ANOVA test.

**Figure 3 ijms-25-10093-f003:**
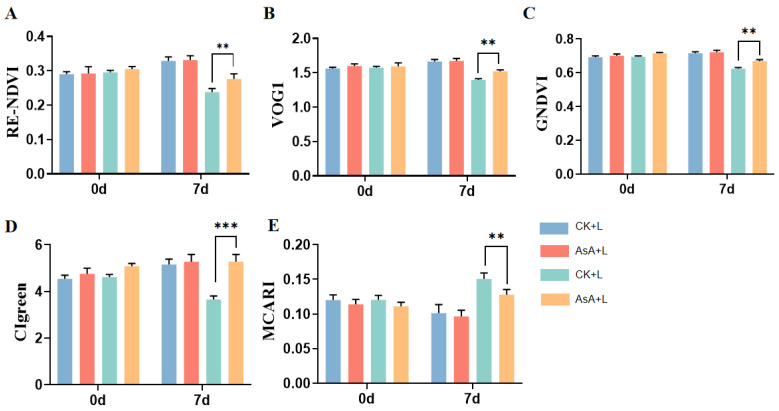
Hyperspectral data of tomato seedlings in each treatment group under cold treatment. Five vegetation indices which were closely related to plant physiology were calculated: (**A**) RE-NDVI. (**B**) SGNDVI. (**C**) VOG1. (**D**) CIgreen. (**E**) MCARI. Data are presented as the means of 3 biological replicates; ** and *** represent significant at 0.01 and 0.001 level, respectively. Using one-way ANOVA test.

**Figure 4 ijms-25-10093-f004:**
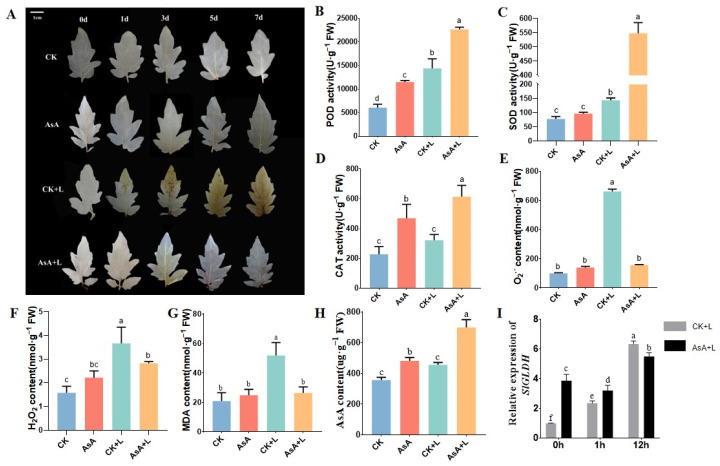
Exogenous AsA enhanced the antioxidant capacity of tomato seedlings under LT. (**A**) Phenotypes of DAB staining. (**B**) POD activity. (**C**) SOD activity. (**D**) CAT activity. (**E**) O^2−^ content. (**F**) H_2_O_2_ content. (**G**) MDA content. (**H**) Endogenous AsA level. (**I**) *SlGLDH* expression level. Data are presented as the means of 3 biological replicates; different letters indicate significant difference at *p*-value < 0.05 using one-way ANOVA test.

**Figure 5 ijms-25-10093-f005:**
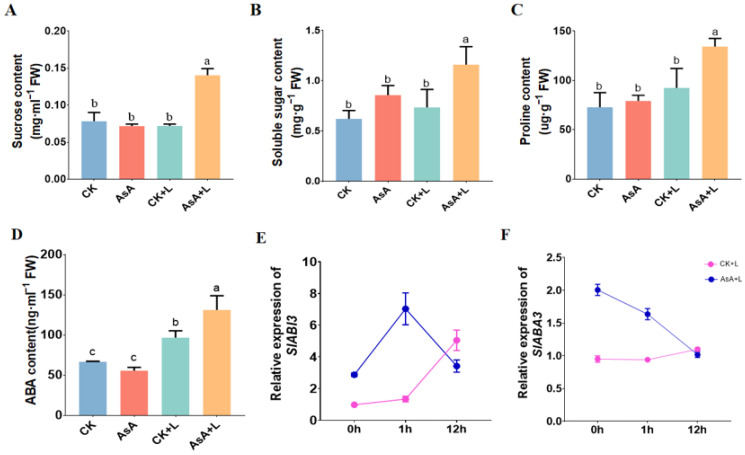
Exogenous AsA increased the accumulation of soluble sugars, proline, and ABA in tomato seedlings under LT. (**A**) Soluble sugar content. (**B**) Sucrose content. (**C**) Proline content. (**D**) Endogenous ABA content. (**E**) *SlABI3* expression level. (**F**) *SlABA3* expression level. Data are presented as the means of 3 biological replicates; different letters indicate significant difference at *p*-value < 0.05 using one-way ANOVA test.

**Figure 6 ijms-25-10093-f006:**
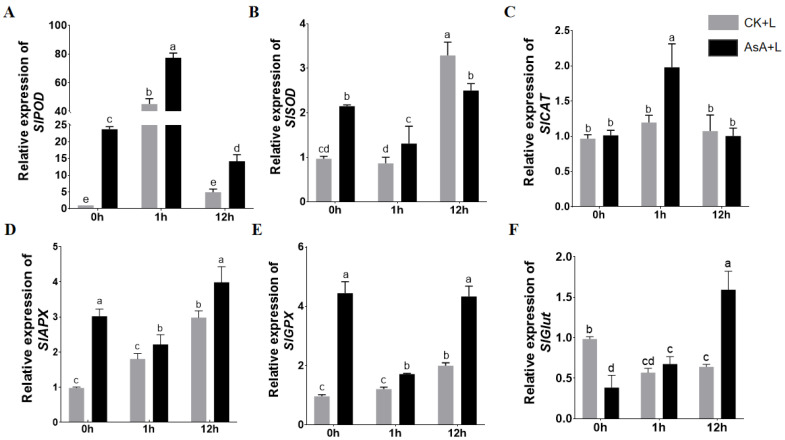
Effect of exogenous AsA on expression levels of antioxidative pathway genes in tomato seedlings under LT. (**A**) *SlPOD* expression level. (**B**) *SlSOD* expression level. (**C**) *SlCAT* expression level. (**D**) *SlAPX* expression level. (**E**) *SlGPX* expression level. (**F**) *SlGlut* expression level. Data are presented as the means of 3 biological replicates; different letters indicate significant difference at *p*-value < 0.05 using one-way ANOVA test.

**Figure 7 ijms-25-10093-f007:**
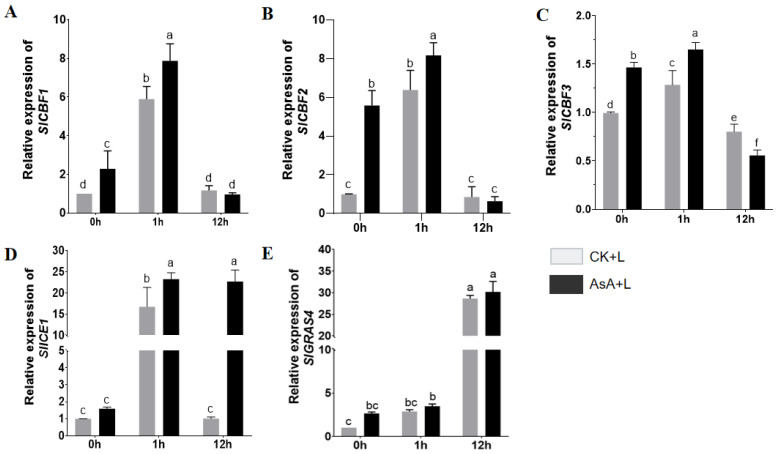
Effects of exogenous AsA treatment on the expression level of cold pathway regulatory genes in tomato seedlings under LT. (**A**) *SlCBF1* expression level. (**B**) *SlCBF2* expression level. (**C**) *SlCBF3* expression level. (**D**) *SlICE1* expression level. (**E**) *SlGRAS4* expression level. Data are presented as the means of 3 biological replicates; different letters indicate significant difference at *p*-value < 0.05 using one-way ANOVA test.

**Figure 8 ijms-25-10093-f008:**
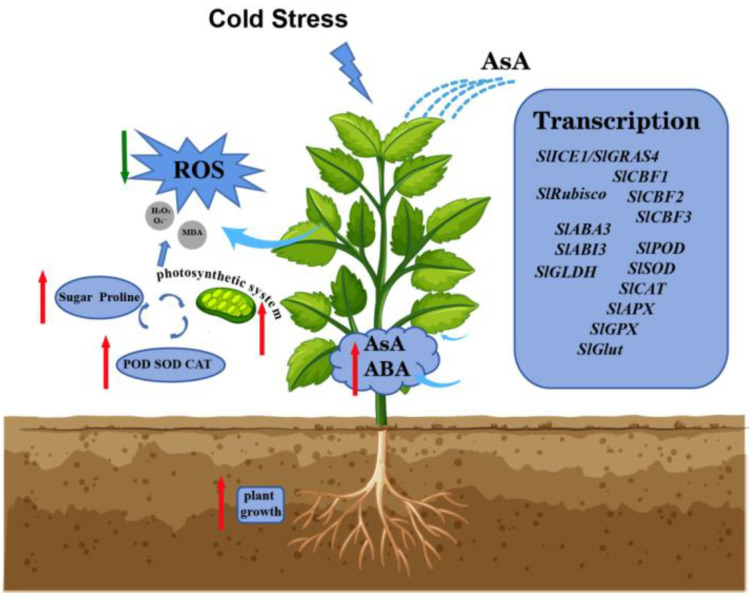
Schematic diagram of exogenous AsA treatment to improve cold tolerance of tomato seedlings under LT.

## Data Availability

Data is contained within the article or Appendix A.

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
