# Peer review of "Application of Exogenous Ascorbic Acid Enhances Cold Tolerance in Tomato Seedlings through Molecular and Physiological Responses"

_ijms, 2024, doi:10.3390/ijms251810093_

Round 1

Reviewer 1 Report

Comments and Suggestions for Authors

The manuscript is well-written. With some minor revisions, the manuscript is acceptable to publish in International Journal of Molecular Sciences journal.

Here are some minor comments to improve.

1.The description of the legend is too simplistic, and the error analysis in the figure should be described in detail.

2.There is a lot of content in the discussion, and it is recommended to reduce it appropriately.

3. All genes should be in italics, please check the whole manuscript.

4. DAB Concentration: Specify the concentration of DAB used in the staining process, as this can significantly affect the results. (lines 311-318)

5. Measurement Conditions: State the specific conditions (e.g., light intensity during measurement) under which chlorophyll fluorescence was measured to ensure reproducibility. (lines 304-310)

6. Ensure that all references cited in this section (e.g., Chen et al., Lu et al.) are included in the reference list and formatted consistently (lines: 284; 307).

7. If Figures or Tables are referenced (e.g., Figure 4B-D and Figure 5B-D), ensure that they are clearly labeled and that their significance is explained thoroughly in the text. Consider adding a brief description of what these figures demonstrate .

Author Response

Comments and Suggestions for Authors

The manuscript is well-written. With some minor revisions, the manuscript is acceptable to publish in International Journal of Molecular Sciences journal.

Here are some minor comments to improve.

1.The description of the legend is too simplistic, and the error analysis in the figure should be described in detail.

Re: Thanks for this considerable suggest, we have improved the description of the legend. Added the error analysis in the figure (figrue1-7).

2.There is a lot of content in the discussion, and it is recommended to reduce it appropriately.

Re: Many thanks for your kind suggestion, we have revised the discussion and streamlined it appropriately.

  1. All genes should be in italics, please check the whole manuscript.

Re: Many thanks for your kind suggestion, we also have checked the whole manuscript and revised this paper carefully.

  1. DAB Concentration: Specify the concentration of DAB used in the staining process, as this can significantly affect the results. (lines 311-318)

Re: Thanks for this constructive suggest, We have use DAB with with a solution concentration of 1 mg/ml and a pH of 3.8, we added (lines 334-335)

  1. Measurement Conditions: State the specific conditions (e.g., light intensity during measurement) under which chlorophyll fluorescence was measured to ensure reproducibility. (lines 304-310)

Re: Thanks for this constructive suggest, We have added this part. (lines 347-351)

  1. Ensure that all references cited in this section (e.g., Chen et al., Lu et al.) are included in the reference list and formatted consistently (lines: 284; 307).

Re: Thanks for this constructive suggest, we have revised this section. (line 299,322)

  1. If Figures or Tables are referenced (e.g., Figure 4B-D and Figure 5B-D), ensure that they are clearly labeled and that their significance is explained thoroughly in the text. Consider adding a brief description of what these figures demonstrate .

Re: Many thanks for your kind suggestion, we have revised this section. (line 449,464)

Reviewer 2 Report

Comments and Suggestions for Authors

Introduction. The aim of the study should be indicated at the end of this section.

Results section

a) "Under cold stress, exogenous application of AsA significantly increased root elongation of tomato seedlings compared to the control group". 

b) "However, the AsA+L group exhibited a slower rate of decrease, and the Chl contents were significantly higher than those in the CK+L group"

c) There are more cases like a) and b) in this section.

The increase/decrease levels should be expressed quantitatively. 

Section 4.2 What concentration of ascorbic acid was used? This should be indicated in this section. How was this concentration selected? Why did the authors select a single concentration for the study?

There are some typos and grammatical issues that require correction throughout the manuscript. The corrections should be highlighted in yellow.

Section 4.6 How were the physiological indicators quantified?

Conclusion

Lines 454-456 showed general information. This is not specific to this work.

Line 461. Figures should not be referenced in this section.

In its present form, this section does not reflect the conclusion of the work. For instance, "In conclusion, our study provides a new explanation for why tomato plants respond better to LT stress through complex biological processes and multifaceted mechanism after application exogenous AsA as a cryoprotectant". Which biological process? Which mechanism?  

References

The format of some references is not homogenous (i.e., 6, 24, 43, and others).

Comments on the Quality of English Language

There are some typos and grammatical issues that require correction throughout the manuscript. The corrections should be highlighted in yellow.

Author Response

Comments and Suggestions for Authors

Introduction. The aim of the study should be indicated at the end of this section.

Re:Many thanks for your kind suggestion. We have revised this part in the new version of manuscript. (lines 96-101)

Results section

  1. a) "Under cold stress, exogenous application of AsA significantly increased root elongation of tomato seedlings compared to the control group". 
  2. b) "However, the AsA+L group exhibited a slower rate of decrease, and the Chl contents were significantly higher than those in the CK+L group"
  3. c) There are more cases like a) and b) in this section.

The increase/decrease levels should be expressed quantitatively. 

Re:Thanks for your considerate suggest. We have addressed your comments regarding the need for quantitative data in the results. (lines 114-115,139,164)

Section 4.2 What concentration of ascorbic acid was used? This should be indicated in this section. How was this concentration selected? Why did the authors select a single concentration for the study?

Re:Thanks for your kind suggest. We selected a concentration of 10 mM AsA based on preliminary experiments where we applied various concentrations of AsA during low-temperature treatment. Our observations indicated that 10 mM AsA yielded the most effective results in terms of enhancing cold tolerance in tomato seedlings. Additionally, we have provided a supplemental figure (Figure S1) that illustrates the results of our experiments with different AsA concentrations, highlighting the superior performance of the 10 mM concentration.

There are some typos and grammatical issues that require correction throughout the manuscript. The corrections should be highlighted in yellow.

Re: Many thanks for your kind suggestion. We have revised all of the errors above in the new version and highlighted these in yellow.

of manuscript.

Section 4.6 How were the physiological indicators quantified?

Re: Many thanks for your kind suggestion. We added this part.(lines 321-329)

Conclusion

Lines 454-456 showed general information. This is not specific to this work.

Re: Thanks for your kind suggest. we have rewritten this description. (lines 482-485)

Line 461. Figures should not be referenced in this section.

Re: Thanks for your kind suggest, hopefully, we have fully understood and properly addressed this issues.(ines 464)

In its present form, this section does not reflect the conclusion of the work. For instance, "In conclusion, our study provides a new explanation for why tomato plants respond better to LT stress through complex biological processes and multifaceted mechanism after application exogenous AsA as a cryoprotectant". Which biological process? Which mechanism?  

Re: Many thanks for this constructive suggest. We have revised this portion to provide a clearer and more detailed explanation of the biological processes and mechanisms involved. (lines 490-498)

References

The format of some references is not homogenous (i.e., 6, 24, 43, and others).

Re: Many thanks for your kind suggestion. All references were checked and revised. (i.e., 6, 24, 28-30, 37-38, 42-44, 46, 49-53, 58-59, 62-64)

Reviewer 3 Report

Comments and Suggestions for Authors

The article is written correctly. The authors presented the reactions of tomato seedlings in the early stages of development to the action of ascorbic acid. They demonstrated a positive effect of ascorbic acid on the physiological and molecular characteristics of plants under winter stress conditions. The authors used the appropriate research methods and correctly interpreted their research results. They presented the results in the form of graphs and photos. In the discussion, they compared their research results with those of other authors. At the end, the authors included a summary. However, the work requires some correction. The introduction lacks a research hypothesis. Please add it. The summary is too general, it should be detailed and respond to the research goal and hypothesis. The work can be published in the International Journal of Molecular Sciences after minor corrections.

Comments on the Quality of English Language

The English language is legible.

Author Response

Comments and Suggestions for Authors

The article is written correctly. The authors presented the reactions of tomato seedlings in the early stages of development to the action of ascorbic acid. They demonstrated a positive effect of ascorbic acid on the physiological and molecular characteristics of plants under winter stress conditions. The authors used the appropriate research methods and correctly interpreted their research results. They presented the results in the form of graphs and photos. In the discussion, they compared their research results with those of other authors. At the end, the authors included a summary.

However, the work requires some correction. The introduction lacks a research hypothesis. Please add it. The summary is too general, it should be detailed and respond to the research goal and hypothesis. The work can be published in the International Journal of Molecular Sciences after minor corrections.

Re:Thanks for your encouragement, and particularly your time. We also added this information in the Introduction. (lines 96-101) And we have improved the description of summary. (lines 12-14,16-24)

Round 2

Reviewer 2 Report

Comments and Suggestions for Authors

Section 2.2 does not show quantitative data. 

"The PSII activity decreased in cold-stressed leaves, along with a sustained decline in the fluorescence parameters Fv/Fm, ETR, and qP"

By how much was the decrease?

"Furthermore, we found that exogenous AsA treatment significantly increased the NPQ value in tomato under cold stress".

By how much was the increase?

Same comment for lines 132-135, 157, 176-178, 183-185, and many other paragraphs in the results section.

Comments on the Quality of English Language

no comments

Author Response

Dear Reviewer,

Thank you for your valuable feedback regarding the lack of quantitative data in Section 2.2 of our manuscript. We appreciate your attention to detail and have now incorporated specific numerical values to support our findings throughout the results section.

1.For the statement regarding PSII activity and fluorescence parameters, we have included the following quantitative data(lines 129-131).

2.Regarding the increase in NPQ value due to exogenous AsA treatment,we have added the following quantitative data(line146).

We have also reviewed the relevant lines you mentioned (132-135, 157, 176-178, 183-185) and added appropriate quantitative data throughout the results section to enhance clarity and support our conclusions,highlighted in green. Thank you once again for your constructive comments, which have greatly improved the quality of our manuscript.

Look forward to receiving your decision.

Best Regards,

Xinman Wang 

On behalf of all the authors
